# Improvement Pathways for Urban Land Use Efficiency in the Beijing-Tianjin-Hebei Urban Agglomeration at the County Level: A Context-Dependent DEA Based on the Closest Target

**DOI:** 10.3390/ijerph20054429

**Published:** 2023-03-01

**Authors:** Ye Tian, Jiangfeng Li

**Affiliations:** School of Public Administration, China University of Geosciences, Wuhan 430074, China

**Keywords:** urban land use efficiency, Beijing-Tianjin-Hebei urban agglomeration, improvement pathways, county level

## Abstract

One of the most effective ways to achieve sustainable land use and the regional coordinated development of urban agglomerations lies in improving the urban land use efficiency (ULUE) of both large, medium, and small cities and small towns. However, in previous studies, less attention has been paid to pathways for potential improvement, especially at the county level. The main purpose of this paper is to examine potential improvement paths for the ULUE at the county level in urban agglomerations, while attempting to provide more practical targets for improvement and formulate more reasonable improvement steps for inefficient counties. Therefore, a total of 197 counties in the Beijing-Tianjin-Hebei urban agglomeration (BTHUA) in 2018 were taken as examples to build a context-dependent data envelopment analysis (DEA) model based on the closest target. In addition, by utilizing methods such as the significant difference test and system clustering analysis, the shortest path and steps to achieve efficiency were identified for inefficient counties, and the characteristics of improvement paths at different levels were summarized. Furthermore, improvement pathways were compared for two dimensions: administrative type and region. The results showed that the causes of polarization for ULUE at different levels were mainly reflected in more complex targets to be improved in the middle- and low-level counties than at high levels. Improving environmental and social benefits was essential to achieving efficiency in most inefficient counties, especially at the middle and low levels. The improvement paths for inefficient counties between different administrative types, as well as the prefecture-level cities, were heterogeneous. The results of this study can provide a policy and planning basis for improving urban land use. This study is of practical significance in accelerating the development of urbanization and the promotion of regional coordination and sustainable development.

## 1. Introduction

Cities are centers of economic activity, innovation, and culture in countries and regions [1]. Large-scale urbanization has led to the ongoing expansion of urban construction while also accelerating the development of urban economies. Nevertheless, it has also caused a series of problems, such as the extensive utilization of urban land [2], reductions in cultivated land [3], intensified energy consumption [4], environmental pollution [5], traffic congestion [6], etc. These have threatened the sustainable development of countries and regions, and it has been proven that the improvement of urban land use efficiency (ULUE) is not only a precondition for promoting urban sustainable development but also a way to balance the development of urban economies and sustainable land use [7]. Hence, it is necessary to change the mode from the extensive one to an intensive one during the process of urban land use. Moreover, significant attention should be paid to the integration of the economic, social, and ecological benefits of land use in order to promote sustainable development for cities [8].

Urban agglomerations are the most dynamic and high-potential regions in China and they play an irreplaceable role in promoting urbanization [9]. In recent years, studies on the ULUE of urban agglomerations have become increasingly popular, including studies on single urban agglomerations such as Beijing-Tianjin-Hebei [10] and the Yangtze River Delta [11], as well as comparative studies of multiple other urban agglomerations [12]. Scholars have mainly emphasized two categories of problems in research on ULUE. The first is referred to as “efficiency evaluation”. Establishing the evaluation index system and determining evaluation methods are the preconditions for evaluating ULUE. With advances in the study of the various conceptions of ULUE, evaluation indexes have evolved from a single index focusing on economic output to a comprehensive index system considering economic, social, and ecological benefits [13]. In the comprehensive index system, the role of environmental factors in ULUE changed from a constraint condition on the quality of economic development to an essential component of comprehensive benefits [14,15]. In terms of evaluation methods, the SFA method has the advantage of having a more explicit economic meaning [16], while data envelopment analysis (DEA) is more suitable for evaluating comprehensive problems with multiple targets [17]. Therefore, the DEA method is used for ULUE by most scholars [18]. The other category is referred to as the “mechanism of efficiency”. Among these approaches, analyzing influencing factors for ULUE helps to understand the driving mechanism behind urban land use [19]. The mechanism research provides guidance for the formulation of macro-level policies. Various influencing factors for ULUE have been addressed extensively in the previous literature [20,21,22].

However, the endogenous differences, meaning equal efficiency scores but significantly different redundant structures in decision making units (DMU), were ignored in the previous literature on mechanism studies. The DEA methods can not only quantify the performance of DMUs, but can also provide improvement benchmarks for inefficient DMUs [23]. Through projection analysis—in other words, comparing the gap between actual and target inputs and outputs—the causes of inefficiency can be identified from a micro-level perspective, and a path to improving efficiency can be determined. The study of the improvement paths for the ULUE is an extension and complement of the above two categories of issues. On the one hand, the analysis of improvement paths further identifies the input and output redundancy of inefficient DMUs based on the efficiency evaluation. On the other hand, compared with the indirect mechanistic analysis of the influencing factors, the analysis of the improvement paths is a direct causal analysis based on efficiency decomposition. However, there are few studies related to efficiency improvement in research on ULUE. Moreover, only a few scholars have conducted empirical analyses at the city level. Fu et al. utilized the slack-based measures (SBM) to evaluate the ULUE of 13 cities in Jiangsu Province and compared the redundancy of undesirable outputs among cities [24]. Han et al. measured the input redundancy of 287 cities in China by constructing an SBM model and revealed the distribution characteristics and regional differences of different redundant factors [25]. Two research perspectives were shown for improvement path analysis in ULUE. One was to analyze which resources are misallocated or wasted from a reasonableness perspective. The other was to identify improvement priorities based on potential targets in economic, social, and environmental benefits from a development perspective, compared to the extent to which different aspects contribute to improving the overall efficiency.

The previous study on improvement paths of ULUE mainly had two deficiencies. One was that the previous ones were too theoretical. More specifically, the practicality of targets for improvement and the actual implementation ability of the research objects was often neglected. The other deficiency was that there were few empirical studies, and the research setting scale was limited. Firstly, from the improvement target perspective, selecting targets that are more in line with experience is a precondition for determining potential pathways for improvement. The improvement targets for inefficient DMUs depend on the distance function of different DEA models [26]. In research on ULUE, DEA models such as Charnes, Cooper, and Rhodes (CCR), slack-based measures (SBM), and their extended models were often used by scholars [27,28]. However, the potential improvements of inefficient DMUs might be overestimated by the traditional DEA model based on the “furthest” targets; meanwhile, the improvement targets obtained in this manner may not be practical targets for inefficient DMUs [29]. Some scholars have pointed out these problems and proposed using DEA models based on the closest target. Minimum distance to a strong efficient frontier (MinDS) is a non-radial DEA model based on the closest target, proposed by Aparicio [30]. The projected point on the frontier obtained by the MinDS model is the nearest efficient projection, meaning achieving efficiency with less effort, to the inefficient DMUs. At present, the closest-target method has been applied to many research fields, such as carbon emission efficiency [31], port efficiency [26], and financial efficiency [32], but, so far, not ULUE.

From the improvement steps perspective, there are certain limitations in analyzing the improvement paths for ULUE utilizing single-frontier DEA models. Previous studies have pointed out that the ULUE of urban agglomerations in China is generally low, and there is an obvious polarization phenomenon [8,33]. In reality, it is difficult to significantly reduce the input or increase the output in the short term. Therefore, the empirical results obtained by using traditional DEA models lack practical significance. Despite the fact that the closest-target DEA model can identify the nearest improvement targets, it may still be difficult to achieve efficiency in one step for inefficient DMUs at short-term time scales. The context-dependent DEA model was proposed by Seiford and Zhu [34]. This model identifies all evaluated DMUs at different layers, which can be seen as multiple frontiers. The efficient projection of inefficient DMUs on the top frontier can be obtained as the ultimate targets, which can be seen as intermediate targets on other frontiers [35]. Thus, the context-dependent DEA model is based on the closest target (context-dependent MinDS, CD-MinDS), which is conducive to exploring a more reasonable and feasible improvement pathway for inefficient DMUs.

In addition, the development strategy of new-type urbanization has been advanced in China since 2012, which pursues the coordinated development of large, medium, and small cities and small towns based on the context of urban agglomerations. However, most existing ULUE studies have been carried out at the city level, mainly focusing on the performance of urban areas while neglecting the evaluation of surrounding county towns. County towns are essential parts of China’s urban system and necessary spaces for promoting industrialization and urbanization [36]. Therefore, it is necessary to take county towns into account in the ULUE study of urban agglomerations. In addition, urban areas were generally evaluated as a whole in research carried out at the city level. In recent years, however, “city–county mergers” have become the primary way for the government to advance urbanization [37]. There are many differences between inner cities and suburbs regarding functional orientation, development levels, and so on. Hence, the ULUE study of urban agglomerations at the county level is helpful in understanding the features and differences of various administrative units, such as inner cities, suburbs, county-level cities, and county towns. From an epistemological perspective, county-level ULUE studies can help to understand more comprehensively and accurately the characteristics and differences in the relative efficiency of different territorial units within urban agglomerations. From the perspective of the ULUE mechanism study, taking county units as research objects can help to further reveal the differences in the causes of different administrative types of inefficient counties in urban agglomerations. The problems of the extensive use of urban land and unbalanced regional development have been still faced by different regions in the Beijing-Tianjin-Hebei urban agglomeration (BTHUA), which is the epitome of China’s urban agglomerations at this point [36]. Therefore, as a research context, the BTHUA is very typical.

In summary, this paper selected 197 counties in the BTHUA as a case study. It utilized the context-dependent DEA based on the closest target to identify the improvement paths for the ULUE. This paper intended to answer the following research questions: (1) Could using context-dependent DEA based on the closest target provide a more practical and reasonable improvement path for inefficient counties than traditional DEA methods? (2) What were the main characteristics of inefficient counties with different levels of efficiency in terms of improvement paths and the impact of economic, social, and environmental benefits on improving efficiency? (3) Were there differences in the improvement paths for inefficient counties by administrative types and regions, and how can we improve the efficiency of different inefficient counties? This paper, by supplementing the existing research methods on urban agglomeration ULUE, adds to the body of empirical research conducted at the county level, thereby assisting China and other developing countries in promoting sustainable urbanization and regionally coordinated development.

## 2. Materials and Methods

### 2.1. Study Area

The BTHUA, located in the northern part of the North China Plain (36°03′–42°40′ N, 113°27′–119°50′ E), includes two municipalities, Beijing and Tianjin, as well as the other cities in Hebei Province. According to the China Statistical Yearbook, in 2018, the urban population of the BTHUA reached 74.24 million, accounting for 5.32% of the country’s total population. The gross domestic product (GDP) was 8513.989 billion yuan, making up 9.47% of the national GDP. The urban construction land area was around 3 million hectares, accounting for 7.63% of the total construction land area of the country. China currently has four primary administrative levels: national, provincial, prefecture-level city, and county. The county-level administrative districts, including counties, districts, and county-level cities, are included in the prefecture-level city’s administrative area. In this paper, the districts are further divided into inner cities and suburbs. A total of 197 counties in the BTHUA were selected as research subjects, including 36 inner cities, 43 suburbs, 21 county-level cities, and 97 county towns. There are only two types of counties in Beijing and Tianjin, with 6 inner cities and 10 suburbs, respectively. Hebei Province has a total of 165 counties, including 24 inner cities, 23 suburbs, 21 county-level cities, and 97 county towns.

### 2.2. Index System and Data Sources

The index system proposed in this paper aims to reflect the relation between input and output in urban economic and social activities, as well as the role of urban land as a geographical space. At present, China’s urbanization has evolved from a period of rapid growth to high-quality development. Therefore, significant attention must be paid to the coordinated development of economic, social, and environmental benefits for urban land use. In particular, not only the level of economic output but also the social welfare and environmental quality should be considered in the examination of efficiency [38]. Therefore, as shown in Table 1, the labor input was expressed as the number of employees per land area and the capital input was expressed as the amount of the fixed asset investment per land area in this paper. In terms of the output of economic benefits, the development level of urban productivity was reflected by the added value of the secondary and tertiary industries per land area. In terms of social benefits, the living and consumption level of residents was expressed by the total retail sales of consumer goods per land area. In addition, the public service level was reflected by the density of points of interest (POI), i.e., the number of POIs per land area [39]. These include four types of data—namely, science and education cultural services, medical care services, transportation facilities and services, government agencies, and social organizations. In terms of environmental benefits, as the BTHUA is among the regions with the most serious air pollution problems in China, air quality improvement is an important goal of high-quality development in this region. In this paper, the annual average concentration of particulate matter (PM2.5) was taken as the undesirable output to reflect the level of environmental quality [40].

Among the relevant data, data on employees, fixed asset investment, the added value of secondary and tertiary industries, the per capita disposable income of urban residents, and the BGR were derived from the 2019 Beijing Area Statistical Yearbook, Tianjin Statistical Yearbook, and the statistical yearbooks of all prefecture-level cities in Hebei Province.

### 2.3. Research Methods

#### 2.3.1. DEA

To explore the influence of different improvement targets and steps on efficiency improvement, several DEA methods were employed in this paper to measure the same sample. First of all, in order to measure the influence of different improvement targets, the SBM model and MinDS model was used to calculate an efficiency score, which served as the basis for the identification of improvement targets. Second, the CD-MinDS model was used to identify the efficiency levels for inefficient counties and measure the gap between inefficient counties and improvement targets, including ultimate and intermediate targets.

SBM and its extended models are among the most widely used DEA models in ULUE research. This model, proposed by Tone et al. in 2001, measures DMU efficiency by the slack of input and output [41]. The model works as follows:(1)$$\min\rho =\frac{1-\frac{1}{m}\sum_{i=1}^{m}\frac{s_{i}^{-}}{x_{ik}}}{1+\frac{1}{q+g}(\sum_{r=1}^{q}\frac{s_{r}^{+}}{y_{rk}}+\sum_{h=1}^{g}\frac{s_{h}^{-}}{z_{hk}})}\;s.t.\left\{\begin{array}{c} \sum \lambda_{j}x_{ij}+s_{i}^{-}=x_{ik}\\ \sum \lambda_{j}y_{rj}+s_{r}^{+}=y_{rk}\\ \sum \lambda_{j}z_{hj}+s_{h}^{-}=z_{hk}\\ \lambda_{j},\;s_{i}^{-},s_{r}^{+}\;,s_{h}^{-}\geq 0 \end{array}\right.$$

In this model, ρ represents the efficiency value of DMU. $s_{r}^{+}$, $s_{i}^{-}$ and $s_{h}^{-}$ are the slack of the i input, the r desirable output and the h undesirable output, respectively. $\lambda_{j}$ is the weight variable of the j unit. $x_{ik}$, $y_{rk}$ and $z_{hk}$ are the $DMU_{k}$ input, desirable and undesirable output values of DMUs, respectively. However, the objective function of the SBM model is to minimize the efficiency value ρ, i.e., to maximize the redundancy of the input and output values. From the perspective of the distance function, the projection point of the DMU is the furthest point on the frontier, meaning that the input and output values for DMUs must be adjusted to the greatest extent. This is obviously contrary to the actual needs of the evaluated objects.

To this end, Aparicio et al. propose the MinDS model to improve the practicality of the SBM model [29]. The objective function of the MinDS model is to maximize the efficiency value ρ by increasing the mixed-integer linear constraint, with the effective DMUs as the reference set and confined to the same hyperplane. The model is described as follows:(2)$$\max\rho =\frac{1-\frac{1}{m}\sum_{i=1}^{m}\frac{s_{i}^{-}}{x_{ik}}}{1+\frac{1}{q+g}(\sum_{r=1}^{q}\frac{s_{r}^{+}}{y_{rk}}+\sum_{h=1}^{g}\frac{s_{h}^{-}}{z_{hk}})}\;s.t.\left\{\begin{array}{c} \vdots \\ -\overset{m}{\underset{i=1}{\sum }}v_{i}x_{ij}+\overset{q}{\underset{r=1}{\sum }}\mu_{r}y_{rj}-\overset{g}{\underset{h=1}{\sum }}\tau_{h}z_{hj}+d_{j}=0\\ v_{i},\mu_{r},\tau_{h}\geq 1\;\\ d_{j}\leq Mb_{j},\lambda_{j}\leq M\left(1-b_{j}\right),b_{j}\in \left\{0,1\right\}\;,j\in E \end{array}\right.$$

The constraint conditions of the MinDS model consist of three parts, among which the first part is the same as the constraint conditions of the SBM model. The common purpose of the second and third parts is to ensure that the reference rods lie in the same hyperplane.

In combination with the MinDS model, the context-dependent model proposed by Seiford and Zhu [34] was employed to provide staged intermediate targets for inefficient counties. The reference set $J^{l}=\left\{DMU_{j},j=1,\cdots ,n\right\}$ was defined as the set containing all DMUs. The iterative reference set $J^{l+1}=J^{l}-E^{l}$ was defined such that $E^{l}=\left\{DMU_{k}\in J^{l}|\rho =1\right\}$ was the set of effective units in the reference set $J^{l}$. When l=1, the model was the MinDS model, $E^{l}$ constituted the global frontier, and the units in the set were effective units. When l=2, the new subset $J^{2}$ was taken as the reference set and recalculated. $E^{2}$ constituted the second-level frontier. The units in the set were inefficient units whose efficiency level was lower than that of the effective units, but higher than that of other inefficient units. All DMUs were divided into different sets by circular calculation. The model is described as follows: (3)$$\begin{array}{c} \begin{array}{c} \max\rho^{*}=\frac{1-\frac{1}{m}\sum_{i=1}^{m}\frac{s_{i}^{-}}{x_{ik}}}{1+\frac{1}{q+g}\left(\sum_{r=1}^{q}\frac{s_{r}^{+}}{y_{rk}}+\sum_{h=1}^{g}\frac{s_{h}^{-}}{z_{hk}}\right)}\\ s.t.\left\{\begin{array}{c} \sum_{j\in E^{l}}\lambda_{j}x_{ij}+s_{i}^{-}=x_{ik}\\ \sum_{j\in E^{l}}\lambda_{j}y_{rj}+s_{r}^{+}=y_{rk}\\ \sum_{j\in E^{l}}\lambda_{j}z_{hj}+s_{h}^{-}=z_{hk}\\ \sum_{i=1}^{m}v_{i}x_{ij}+\sum_{r=1}^{q}\mu_{r}y_{rj}-\sum_{h=1}^{g}\tau_{h}z_{hj}+d_{j}=0\\ \lambda_{j}\geq 0\\ s_{i}^{-},s_{r}^{+},s_{h}^{-}\geq 0\\ d_{j}\leq Mb_{j},\lambda_{j}\leq M\left(1-b_{j}\right),b_{j}\in \left\{0,1\right\}\\ E^{l}=\left\{DMU_{k}\in J^{l}|\rho \left(l,k\right)=1\right\}\\ l_{0}\in \left\{2,\cdots ,L\right\}\\ p\in \left\{1,\cdots ,l_{0}-1\right\} \end{array}\right. \end{array} \end{array}$$

In this model, the reciprocal of $\rho^{*}$ was the progress value of $DMU_{k}$ based on $E^{l_{0}-p}$, representing the improvement degree required for $DMU_{k}$ to raise the efficiency level to $E^{l_{0}-p}$,. Here, $1/\rho^{*}>1$, and the higher the $1/\rho^{*}$ value, the greater the improvement degree. When $DMU_{k}$ had multiple superior frontiers, $\rho^{*}\left(p+1\right)<\rho^{*}\left(p\right)$. In addition, the improvement ratio of input and output elements (I/O elements) in $DMU_{k}$ was used to characterize the statistical redundancy for each element—that is, the ratio of the slack of each element to the actual value. The improvement ratios of input, desirable output and undesirable output are as follows:(4)$$\frac{s_{i}^{-}}{x_{ik}},\;\frac{s_{r}^{+}}{y_{rk}},\frac{s_{h}^{-}}{z_{hk}}$$

#### 2.3.2. Paired-Samples *t*-Test and One-Way ANOVA

To determine reasonable improvement targets and steps, the mean difference significance test was used to judge whether the results obtained by the DEA model were statistically different. The paired-samples *t*-test can be used to test whether there is a significant difference in the mean values of the two groups’ paired sample data. As mentioned above, in terms of DEA principles, the shortest improvement paths for inefficient counties can be identified by the closest target. In other words, the MinDS model proposes more practical ways to improve efficiency for inefficient counties, enabling them to become efficient with less effort. To verify the validity of the MinDS model for the ULUE study, the MinDS model was compared with the SBM model, which was widely used to evaluate ULUE. Specifically, suppose that the efficiency scores of the SBM model are significantly lower than that of the MinDS model. In this case, it indicates that the MinDS model can identify the more practical paths for improving ULUE in inefficient counties. Furthermore, the one-way ANOVA test can be used to test for significant differences between multiple sample data. Therefore, the improvement steps of inefficient counties were determined via a one-way ANOVA test according to the improvement degrees of inefficient counties at different levels. Suppose that the improvement degree significantly differs based on the global frontier in inefficient counties at different levels. In this case, the improvement steps for inefficient counties at low levels are unreasonable, and the intermediate targets should be added to reduce the improvement degrees of each step. Finally, the overall improvement degrees of inefficient counties were compared to test whether there were significant differences between one-step and step-by-step by the paired-samples *t*-test. If the overall improvement degree of the step-by-step is less than that of the one-step, then a more practical and reasonable improvement path can be identified by the CD-MinDS model. Otherwise, the improvement path needs to be weighed between the one-step and step-by-step.

#### 2.3.3. System Clustering Analysis

As an exploratory method, system clustering analysis can be divided into variable clustering (Mode R) and sample clustering (Mode Q). The method classifies objects with similar properties according to the degree of closeness between variables or samples. In this paper, Mode Q systematic clustering was used to identify the similarity of redundant elements among inefficient counties based on the adjusted cosine similarity, and this was used to classify the key elements for improvement at each stage.

## 3. Results

### 3.1. Efficiency of the Closest and Furthest Targets

In order to compare the influence of different improvement targets on inefficient counties, the efficiency scores based on the furthest target (the SBM model) and the closest target (the MinDS model) were measured according to Formulas (1) and (2), and the results were compared by using the paired-samples *t*-test. The most effective and ineffective counties obtained by the two models were the same, 26 and 171, respectively. As shown in Table 2, the efficiency scores based on the nearest target were significantly higher than those of the furthest target (*p* < 0.01), and there was a significant correlation between them (*p* < 0.01). This shows that the influence of different measurement benchmarks was clear when both the frontier and inefficient counties were the same in the two groups of samples. For inefficient counties, taking the projection identified by the nearest-target method as the improvement goal can achieve the same effect with relatively small improvements.

According to Formula (3), the inefficient counties were stratified to further distinguish the efficiency level among inefficient counties. As shown in Figure 1, the counties of the BTHUA were divided into seven levels. The counties located at the frontier of the first level (global frontier) were effective counties, and those located at the frontier (local frontier) of the 2nd–7th levels were inefficient counties. The efficiency level of counties at the same level was the same, and showed a decreasing trend from first to seventh. The number of counties in the 2nd–5th levels was relatively large (30–35), while that in the seventh level was relatively small (13). From the regional perspective, the counties of Beijing and Tianjin occupied most of the counties in the first level. Most of the counties in the 3rd–7th levels were in Hebei. This indicates that the efficiency levels of Beijing and Tianjin counties in the BTHUA were similar, and the efficiency performance was better. In contrast, the efficiency levels of counties in Hebei Province ranged more widely and the efficiency performance was relatively poor.

### 3.2. Intermediate Targets and Steps for Improvement

In order to compare the improvement degrees for inefficient counties at different levels, the progress values based on the global frontier were calculated according to Formula (3). This was done in order to characterize the improvement degrees required to reach the efficient level. The results were then compared by using ANOVA. As shown in Table 3, the Levene statistic was 1.709, with a significance level of over 0.05, meeting the requirements of ANOVA. The F statistic for the ANOVA test was 7.524, with a significance level of less than 0.05, indicating that there were significant differences in the average improvement degrees of the counties at various levels. Through multiple comparisons among different levels of counties (Figure 2), it was found that there were significant gradient differences in the improvement degrees for counties at different levels. There were significant differences among high-level (the second level), middle-level (the third and fourth levels), and low-level (the 5th–7th levels) counties, but there were no significant differences within the groups. The improvement degree was the highest in low-level counties, followed by middle-level counties and then high-level counties. This shows that the improvement process for counties at all levels was asynchronous, and it was obviously longer in middle and low-level counties. Therefore, it is necessary to further set intermediate targets and formulate progressive improvement steps.

The intermediate targets are determined according to the differences in their improvement degrees. In terms of the 5th–7th levels of counties, the improvement degrees based on the global frontier were significantly greater than that of the high- and middle-level counties, with the smallest difference for the fourth level of counties. Therefore, the fourth level can be regarded as the intermediate target of the first step, and the second level can be regarded as the intermediate target of the second step, dividing the improvement process into three steps. Similarly, the frontiers for the second level are taken as the intermediate targets for counties at the 3rd–4th levels. The improvement processes for three groups of counties at high, middle and low levels have one, two and three steps, respectively. Table 4 presents the inspection results based on intermediate targets. It should be noted that the improvement degrees for the second and third steps were calculated by taking the target input and output of the previous step as the actual input and output of this step. It can be seen from the results that there was no significant difference in terms of the improvement degrees of counties at all levels during the three steps, and the average improvement degree in the three steps was relatively low. This means that the improvement process for middle- and low-level counties can be decomposed by setting intermediate targets so that the improvement degree of each step is in a more reasonable range. Therefore, it is appropriate to take the frontiers of the second and fourth levels as the intermediate targets of the middle- and low-level counties.

To further explore the reasons that there were various improvement degrees, the redundancy (improvement ratio) of elements of input and output in inefficient counties was calculated based on Formula (4) and the redundant quantity in each county was counted. Figure 3 presents the statistical redundancy of the elements of input and output in the case of one-step and step-by-step in high-, middle- and low-level counties, as well as the proportions of different redundant quantities. On the whole, middle- and low-level counties had larger redundant quantities and greater redundancy, indicating that there are more aspects to be improved in middle- and low-level counties, with more difficulties during the improvement. The establishment of intermediate targets plays a role in screening and focusing on the improvement elements. In other words, when the local frontiers that are more similar to themselves are considered as benchmarks, there are smaller quantities and lower redundancy. Therefore, setting intermediate targets can help to recognize the main weaknesses of each stage, and improvement in these aspects may be a shorter path to advance efficiency.

In addition, paired-samples *t*-tests were used to compare the progress values for the two groups to further explore the overall improvement degrees for both one-step and step-by-step. The slack of the elements of input and output during the step-by-step method was the sum of the slack in each stage. As shown in Table 5, there was a significant difference between the progress values for the two groups (*p* < 0.01). The progress value in terms of step-by-step was considerably lower than that of the one-step context, showing a significant correlation between the two (*p* < 0.01). Thus, the method of step-by-step is a shorter path for middle- and low-level counties to achieve efficiency using a step-by-step method, which supports the conclusion above.

### 3.3. Improvement Elements of Inefficient Counties

In this paper, the Q-type system clustering method was used to classify the input and output redundancy of inefficient counties for exploring the characteristics of improvement elements at each stage. In Figure 4, the six improvement elements in the first step are shown. On the whole, one improvement element had the most prominent improvement ratio in each type, which was significantly higher than that of other redundancies, indicating the existence of one major improvement element. From the perspective of the improvement elements, there were mainly deficiencies in economic, social, and environmental benefits, reflecting the low degree of intensive utilization of urban land in inefficient counties, and the continuation of the extensive land use mode, to some extent. In terms of the number of counties of various types, the number of counties with tertiary industry as the key improvement element was the largest (47), followed by counties with the secondary industry, resident income, AQI or BGR as the major improvement element. It was revealed that the improvement elements in the first step were diverse. The types of improvement elements in the second and third steps were slightly fewer than in the first step, both showing four types (Figure 5). In these two steps, most counties needed to further improve the output of economic and environmental benefits on the basis of the first step to achieve the final goal. This means that the output of economic and environmental benefits in the middle- and low-level counties was generally insufficient, with a large gap compared with the highest level in the urban agglomeration. In terms of the numbers of counties, in the second step, counties taking the BGR as the improvement element were obviously greater in number. In the third step, similarly, the number of counties with the BGR and tertiary industry as the improvement element was larger. This initially reflects that BGR and tertiary need to be focused on both in the middle- and low-level counties.

Combining the improvement elements in different steps, 54 ULUE improvement paths were formed in 171 inefficient counties (Figure 6). The number of paths was the largest in low-level counties, followed by middle-level counties and high-level counties. On the whole, most high-level counties needed to focus on improving economic benefits. In addition to economic benefits, middle- and low-level counties generally needed to promote social or environmental benefits. On the basis of the similarity in terms of improvement elements, 54 improvement paths could be summarized as the economic (Ec.), economic–social (Ec.–Soc.), economic–environmental (Ec.–Env.), social–environmental (Soc.–Env.) and economic–social–environmental (Ec.–Soc.–Env.). The number of counties was the largest for the Ec.–Env. category (68), followed by Ec. (45), Ec.–Soc.–Env. (24), Soc.–Env. (18) and Ec.–Soc. (16). Figure 7 shows the proportions of various improvement elements in high-, middle- and low-level counties from both short-term and long-term perspectives. In the short term, the improvement elements for high-level counties were mainly those related to tertiary industry. The improvement elements for middle- and low-level counties had significant heterogeneity, and the proportion of each element did not exceed 25%. In the long run, the middle- and low-level counties were characterized by great differences in terms of the social and environmental domains. The middle-level counties had a more significant direction of improving efficiency in the social context, while the environmental domain was more critical for low-level counties as key improvement elements.

### 3.4. Improvement Paths of Different Types and Regions

In this section, the improvement paths of the inefficiency units were compared further from the dimensions of administrative types and regions (prefecture-level cities). Figure 8 presents the proportions of high-, middle- and low-level counties that are different types and located in regions. From the perspective of administrative types, most of the inner cities, suburbs and county-level cities were high-level and middle-level counties, while the county towns were mostly low-level counties. In terms of regions, the difference was obvious between cities; however, it was not significant within the cities. There were mainly high-level counties in Beijing and Tianjin. Four cities in Hebei along Beijing and Tianjin, and the Langfang–Tangshan–Qinhuangdao Axis, were mainly middle-level counties. The majority of the six cities in the central south were low-level counties. The results above mean that, in terms of improvement steps, the regional difference is more prominent, showing a pattern of difference between the north and south. The urban land use and management may have boundary effects—namely, the urban land use levels of various counties within the city area are relatively similar.

Figure 9 demonstrates the differences in improvement elements in terms of different types of settlements and regions. From the short-term perspective, the proportions of improvement elements of the four types of counties were all less than 35%. Furthermore, the improvement elements with the highest proportion are in inner cities and suburbs, which belong to tertiary industry. In contrast, the highest proportion of elements in county-level cities and county towns are public services and air quality. This phenomenon also took place within the cities where improvement elements presented obvious heterogeneity, as well. From a longer-term perspective, the improvement elements for county-level cities and county towns were relatively concentrated, similar to the seven cities in Hebei. Specifically, county-level cities and county towns were mainly associated with the economic and environmental contexts, which is the same as four cites—the traditional industrial city. In addition, two cities in Northern Hebei, Chengde, and Zhangjiakou, were associated mainly with the social context in Northern Hebei. The traditional industrial city in Southern Hebei, as well as Xingtai and Handan, were mainly associated with the economic–social–environmental contexts. It is evident that the functional layout of each region within cities is not currently reasonable, and some portions of cities have common problems in terms of the process of efficiency improvement, which needs to be advanced in general.

In this paper, cross-analyses from the dimensions of administrative types and cities were conducted and presented in the form of a matrix heat chart (Figure 10) in order to further examine whether there were common features in the short-term improvement elements of inefficient counties belonging to different types and located in various regions. In Figure 10, there are 52 regional units composed of 4 administrative types and 13 cities. This concentration reflects the proportion of the dominant improvement elements in the unit, i.e., the ratio of the number of counties with that improvement element to the total number of counties in the unit. It can be seen that 16 region units shared consistency in short-term improvement elements, accounting for 30.77%, distributed in seven cities. According to the number of these units, the improvement element was dominated by air quality, followed by green space, public services, secondary industry and resident income. The results demonstrate that a small number of regions in the BTHUA have common features in the short-term improvement elements, with a relatively scattered distribution.

## 4. Discussion

### 4.1. Theoretical Implications

The DEA method has been widely adopted in ULUE research. In this paper, improvement paths for inefficient counties in the BTHUA were analyzed by combining the nearest target and context-dependent DEA. The results show that, with the same technology frontier and in the same inefficient counties, selecting different benchmarks as improvement targets can have a significant impact on redundant information. From the perspective of efficiency improvement, it is a more desirable choice for inefficient counties to achieve the same effect with relatively small adjustments. In previous studies, the SBM model was mainly used to evaluate ULUE [42]. However, the SBM model maximized the redundancy of input and output, thus leading to an underestimation of efficiency. This paper has compared the results based on the nearest target (MinDS) and the furthest target (SBM) by utilizing paired-samples *t*-tests, indicating that the efficiency of MinDS was significantly greater than that of SBM. This means that identifying the redundant information of inefficiency units through the context-dependent DEA based on the closest target helps to find shorter improvement paths. Moreover, the principle of the nearest target method is to take the projection of the most similar actual input and output of the inefficiency county as the evaluation benchmark. Therefore, the redundant information and improvement targets at the basis of the method are more practical and instructive.

Adding intermediate targets is helpful in providing shorter improvement paths for middle- and low-level counties and the improvement degrees, keeping them within a reasonable range at different stages. With the CD-MinDS model, improvement degrees for inefficient counties were calculated and compared under the conditions of both the single benchmark (one-step) and the multi-level benchmark (step-by-step). According to our results, improvement degrees during the step-by-step improvement were significantly lower than those during the one-step. The reason is that the inefficient counties in urban agglomerations have high heterogeneity in terms of land use modes, economic and social development levels, etc. The number of counties constituting the global frontier is small and relatively homogeneous in type. The evaluation results based on a single benchmark may be prone to miscalculation due to heterogeneity, leading to the overestimation of potential improvements. As the types of counties constituting the local frontier are more diverse and more similar to the evaluated counties, the miscalculation caused by heterogeneity is reduced to some extent. In addition, for middle- and low-level counties, the improvement information provided by a single benchmark is too general, while the intermediate targets can serve as a guide and help to better understand the key improvement elements at each step.

In this study, the improvement paths of ULUE were analyzed for the BTHUA in 2018 at the county level. The results showed that, among the 197 counties, there were 184 inefficient counties, including 21 high-level counties, 59 middle-level counties and 104 low-level counties. Significant gradient differences were evident in the improvement degrees for high-, middle- and low-level counties. There was also a large gap between the efficiency level for most counties and the highest efficiency level of the urban agglomeration. Within the urban agglomeration, there was a spatial non-equilibrium characteristic of polarization. In other words, the efficiency level for each county in the core city was the highest and the differences were considerable between surrounding cities versus within the cities. This is again consistent with an overall pattern of high levels in the north and low levels in the south. The results were similar to those of other studies on other urban agglomerations in China, such as those in the Yangtze River Delta [11], the Pearl River Delta [42] and the Shandong Peninsula [43]. These results are in accordance with the findings of Fang et al. that the urban agglomerations in China are still in the initial stage of development or the fast-growing stage, emphasizing the fact that the sustainable development of urban agglomerations should follow an agglomerated effect strategy and borrowed size [44]. In addition, compared to the existing study, the causes of efficiency differences were further analyzed by identifying redundancy characteristics in inefficient counties at different levels as improvement elements. The improvement elements of high-, middle- and low-level counties were classified to reveal the direction of improvement of ULUE. The types of improvement elements were more concentrated in the high-level counties, and the economic contexts, especially the tertiary industry, accounted for the majority. The types of improvement elements are more diverse in the middle- and low-level counties compared to the high-level counties. Specifically, the short-term improvement elements in most middle- and low-level counties are social or environmental contexts, and the share of each element is below 25%. Meanwhile, the long-term improvement elements in these counties contain two to three improvement elements, with a high percentage of them containing both economic and environmental contexts, followed by economic, social and environmental or social and environmental. The results indicate that the cause of the polarization of ULUE in the BTHUA is the presence of more weaknesses in the outputs of middle- and low-level counties. On the other hand, environmental and social benefits are the key to improving ULUE in the short term in middle- and low-level counties. They are also necessary conditions to achieve full efficiency.

Industrial growth enhances overall economic strength [21], as well as ULUE. However, increasing the share of services in the industrial structure is considered to be a more general view to promote higher ULUE [45,46]. In addition, stressing the regulation of environmental pollution and increasing public service expenditures also have a significant positive impact on ULUE [47,48]. In this study, however, the improvement paths for inefficient counties were identified from the micro-level perspective. The results showed that obvious heterogeneity was manifested in the improvement elements for inefficient counties with different administrative types and regions. From the short-term perspective, the improvement elements in most inner cities are economic contexts. In contrast, the improvement elements in county-level cities and county towns are mainly social or environmental contexts. Similarly, the improvement elements of inefficient counties in Beijing and Tianjin are mainly economic contexts, while inefficient counties in most of Hebei’s prefecture-level cities are social or environmental. From a long-term perspective, the types of improvement elements in county cities and county-level cities are more concentrated, mostly economic–environmental, while the improvement elements in inner cities and suburbs are mainly economic or economic–environmental. The types of improvement elements of inefficient counties within each prefecture-level city in Hebei are more concentrated, mostly economic–environmental and a few economic. In contrast, the improvement elements in Beijing and Tianjin are more diverse, including economic, economic–social and economic–environmental. The results demonstrate that the causes of ineffectiveness are different across the administrative types and regions of counties and that inefficient counties need to be targeted for improvement according to their critical weaknesses to achieve efficiency. Future ULUE studies on urban agglomerations should take into account the heterogeneous background of the research subjects, while at the same time comprehensively analyzing and discussing the influence mechanisms and efficiency improvement from both macro- and micro-level perspectives.

### 4.2. Policy Implications

In the past decade or so, China has experienced a surge in urbanization. For example, during the period between 2012 and 2018, all 13 prefecture-level cities except Xingtai and Cangzhou in the BTHUA expanded their urban scales by the means of a “city–county merger”. In terms of ULUE, large-scale urbanization has not brought significant efficiency advantages, especially in the Hebei cities. As China’s old industrial bases and resource-based cities, these cities are facing the dilemma of transforming and upgrading their leading industries. In terms of accelerating the restructuring of economies, local governments have guided the transfer of secondary industries, which are mainly labor-intensive and resource-intensive, from the inner cities to suburbs and surrounding counties. This leads high-tech industries and producer services to be the new driving forces for urban development. Through the improvement elements of various counties within the prefecture-level cities, it is evident that, in the inner cities, the secondary and tertiary industries make up major proportions. In the suburbs and county towns, on the other hand, secondary industries and air quality account for more. On the one hand, this reflects the ineffectiveness of the main urban areas in attracting and nurturing emerging industries, as well as the continuation of the extensive urban land use mode in peripheral areas. On the other hand, this reflects a lack of industrial support and collaboration capacity within the cities. Therefore, it is necessary to strengthen the coordinating role of regional governments and to break down the administrative barriers between cities. This allows them to benefit from the other’s comparative advantages through cross-city collaboration, which promotes effective urban land use in each region of the urban agglomeration.

In the BTHUA, inefficient counties belonging to different types and located in various regions have obvious heterogeneity in the improvement degrees and elements, indicating that more precise governance must be implemented. The governance of urban agglomerations needs to focus on adopting policies based on regional and special planning, while adhering to systematic and comprehensive approaches. From an administrative type perspective, the inner city is a multifunctional center within the prefecture-level city limits, generating a substantial economic radiation role in the surrounding areas. The improvement of the inner cities in the BTHUA, mainly in Hebei, was economic output, including secondary and tertiary industries. Inner cities with relatively low efficiency should continue the strategy of industrial transformation. It is vital to devise more positive industrial and land use policies to increase the share of productive services and high-tech industries in the economic output and promote the redevelopment of inefficient urban land. The suburbs bear the function of taking over the industrial transfer from the inner cities and are potentially densely populated areas in the urbanization process. The direction for improving ULUE in the suburbs was mainly economic and environmental, primarily tertiary industry and BGR. The suburbs should adopt the strategy of city–industry integration to provide more attractive talent policies and focus on improving the business environment to improve the local industrial structure. Meanwhile, it is necessary to implement a sustainable urban operation model and expand the size of urban green spaces. County-level cities and county towns are satellite towns closely linked to the central urban areas and regional centers that surround the rural hinterland. Compared with the inner cities and suburbs, most county-level cities and county towns have more improvement elements and steps. A long-term and tailored development plan is essential. For instance, industrial-oriented county towns should adopt a strategy of industrial upgrading and subsidize R&D and innovation for township enterprises to achieve higher-quality economic output, thereby improving local living standards and reducing negative environmental impacts.

### 4.3. Limitations and Future Improvements

This study has certain limitations. First, this is a study on ULUE for a single urban agglomeration, the BTHUA. Considering the diversity and complexity of cities in China, it is necessary to include a wider range of urban agglomerations as samples for research. Second, this paper used cross-sectional data to explore the improvement paths for ULUE at the county level. In the future, introducing panel data will be required to further identify and compare the improvement paths for inefficient counties in the process of dynamic change. Moreover, in terms of the construction of the evaluation index system, in order to explore green use and sustainable development for urban land more thoroughly, data sources must be further broadened in the future by including energy consumption, carbon emissions, etc., in the evaluation and analysis of ULUE. In addition, in terms of research methods, this paper has mainly discussed efficiency improvement for inefficient counties from the micro-level perspective. From the macro-level perspective, however, the improvement of ULUE is also influenced by various exogenous drivers. Therefore, in the future, making further efforts to combine micro- and macro-level perspectives will be necessary to explore the mechanisms and improvement paths for ULUE.

## 5. Conclusions

In this study, the nearest targets and context-dependent DEAs were combined to evaluate and identify ULUE and improvement paths for 197 counties in the BTHUA in 2018. The improvement targets and steps for inefficient counties were compared and analyzed using ANOVA and paired-samples *t*-tests. The improvement elements for inefficient counties were classified and summarized according to the Q-type system clustering method, and improvement paths for inefficient counties belonging to different types and located in various regions were further compared. The main conclusions are as follows. (1) Compared to previous DEA methods for ULUE, better-matched improvement targets are identified by CD-MinDS, resulting in significantly shorter improvement paths. The decomposition of the improvement process by adding intermediate targets helps to identify more reasonable steps and more practical guidance for middle- and low-level counties during the step-by-step method. (2) The inefficient counties in the BTHUA account for more than 85% of the total, and the improvement processes for inefficient counties at high, middle and low levels have one, two and three steps, respectively. The types of improvement elements are more concentrated in the high-level counties and more diverse in the middle- and low-level counties. The economic benefits have a widespread impact on the improvement efficiency of inefficient counties at all levels. However, the environmental and social benefits have a crucial impact on achieving full efficiency for most middle- and low-level counties. (3) The obvious heterogeneity is revealed in the improvement elements for inefficient counties with different administrative types and regions in the short and long term. The inefficient counties should make targeted improvements to achieve efficiency by addressing their critical weaknesses at different stages.

## Figures and Tables

**Figure 1 ijerph-20-04429-f001:**
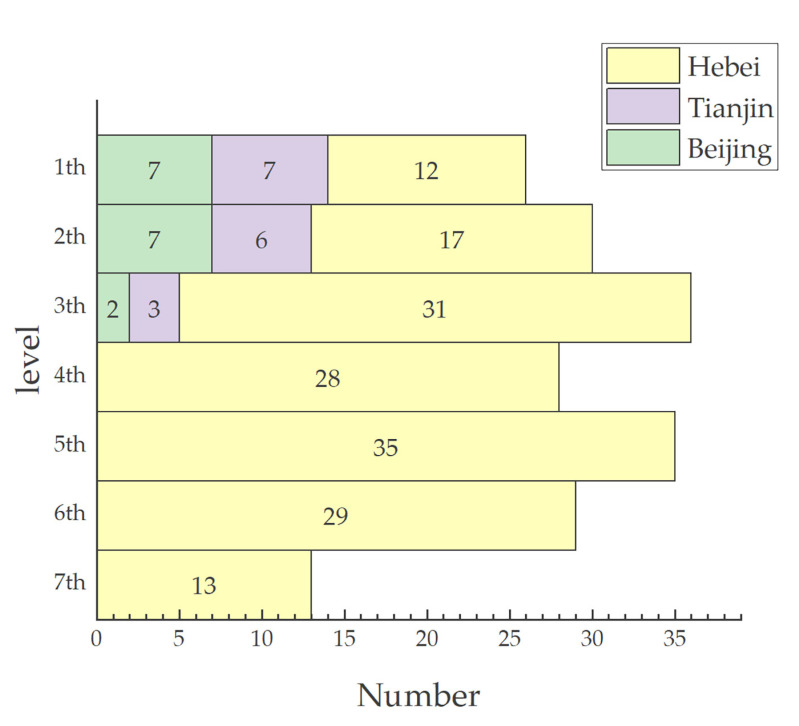
Number and regional distribution of nine efficiency levels in counties.

**Figure 2 ijerph-20-04429-f002:**
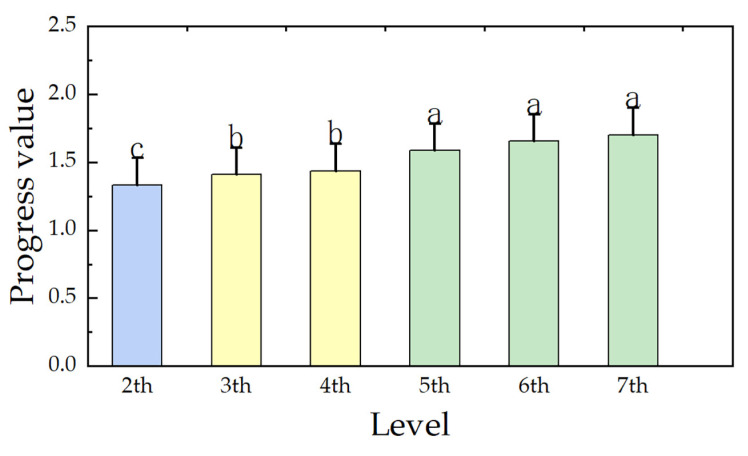
Comparison of global progress values of counties at different levels. Note: Different letters represent a significant difference at 5% level.

**Figure 3 ijerph-20-04429-f003:**
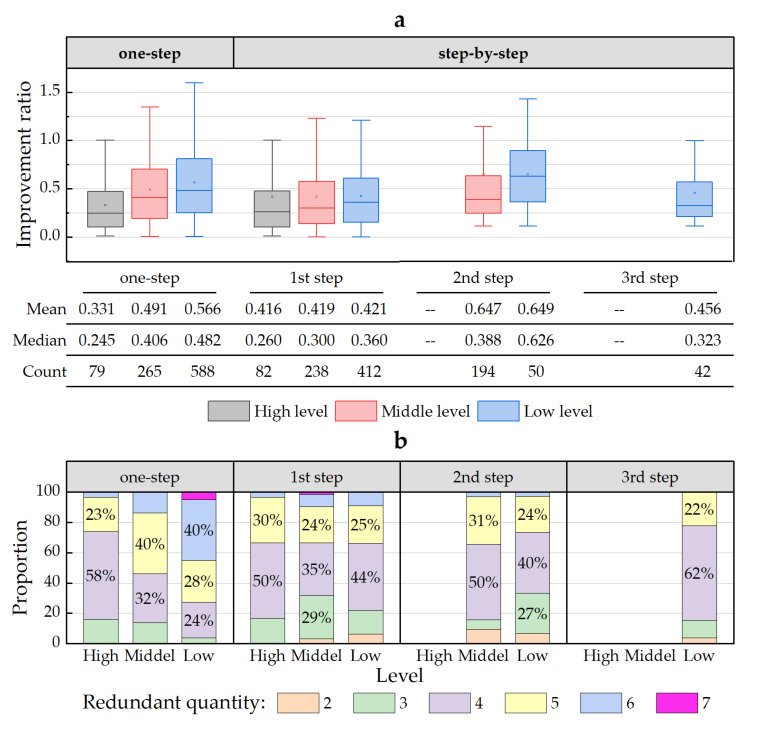
Differences for input and output redundancy among high-, middle- and low-level counties. (**a**): boxplot of redundancy improvement ratio for counties at each level. (**b**): proportion structure of the redundant quantities of counties at each level.

**Figure 4 ijerph-20-04429-f004:**
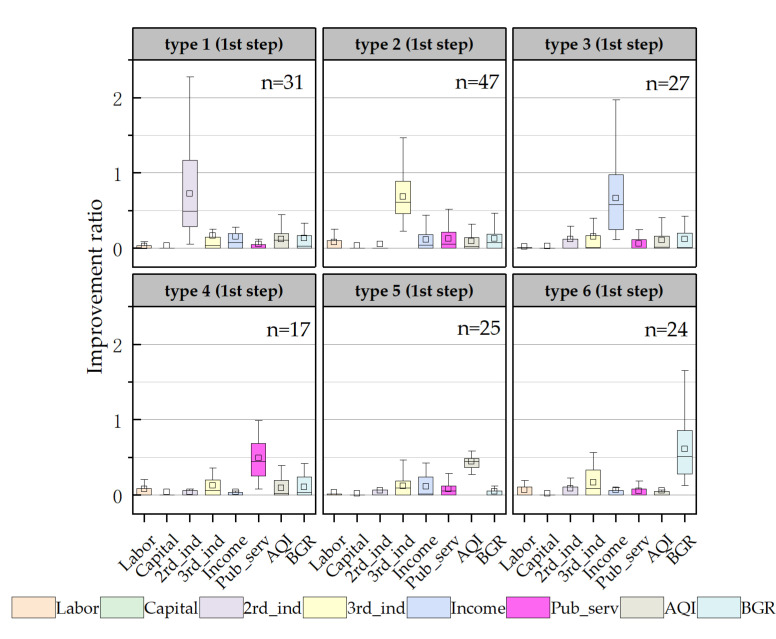
Types of key improvement elements in the first step. Note: 2nd_ind—secondary industry; 3rd_ind—tertiary industry; Income—resident income; Pub_serv—public services; AQI—air quality; BGR—green development level.

**Figure 5 ijerph-20-04429-f005:**
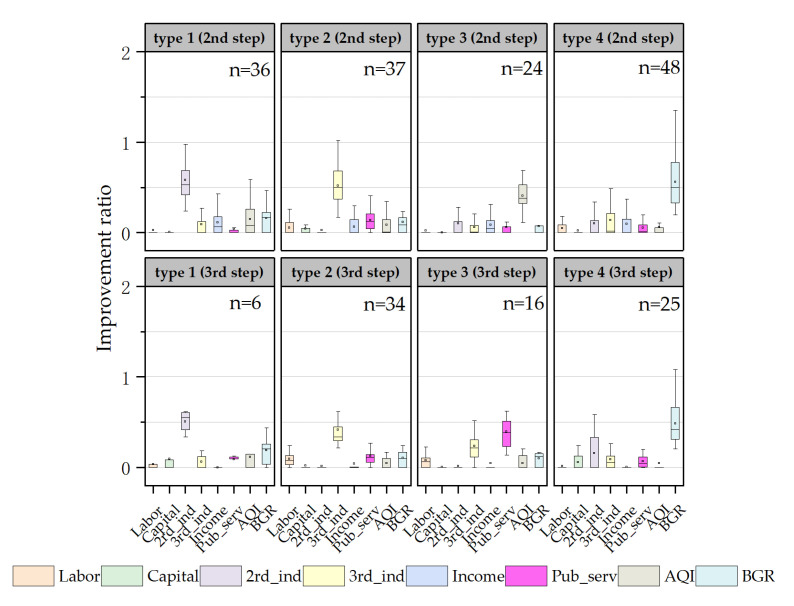
Types of key improvement elements in the second and third steps. Note: 2nd_ind—secondary industry; 3rd_ind—tertiary industry; Income—resident income; Pub_serv—public services; AQI—air quality; BGR—green development level.

**Figure 6 ijerph-20-04429-f006:**
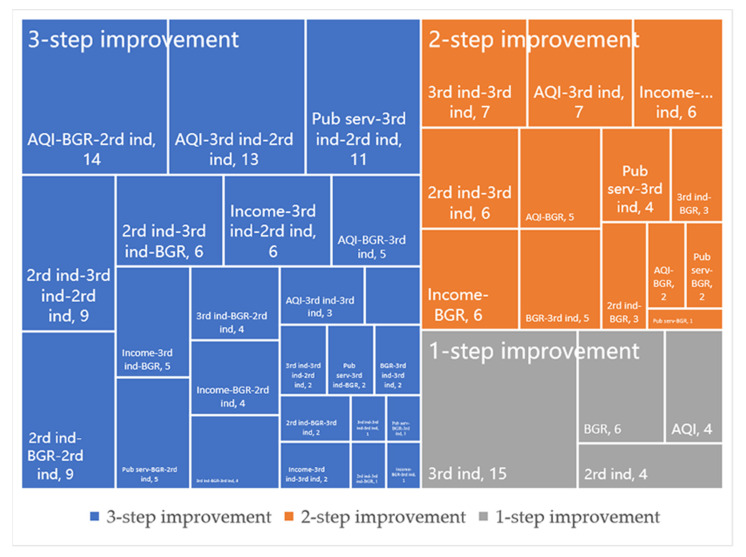
Forty-one improvement paths in inefficient counties. Note: 2nd_ind—secondary industry; 3rd_ind—tertiary industry; Income—resident income; Pub_serv—public services; AQI—air quality; BGR—green development level.

**Figure 7 ijerph-20-04429-f007:**
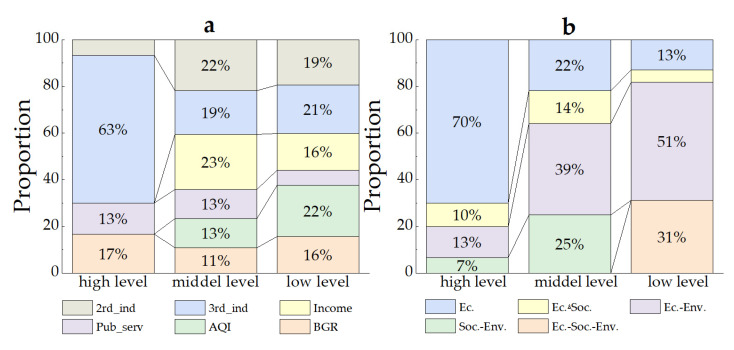
(**a**): Proportions of short-term improvement element of high-, middle- and low-level counties. (**b**): proportions of long-term improvement element of high-, middle- and low-level counties. Note: 2nd_ind—secondary industry; 3rd_ind—tertiary industry; Income—resident income; Pub_serv—public services; AQI—air quality; BGR—green development level.; Ec.—economic; Ec.–Soc.—economic–social; Ec.–Env.—economic–environmental; Soc.–Env.—social–environmental; Ec.–Soc.–Env.—economic–social–environmental.

**Figure 8 ijerph-20-04429-f008:**
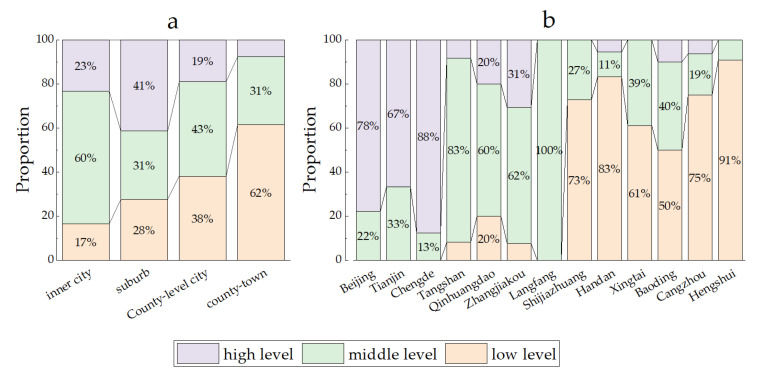
(**a**): Proportions of high-, middle- and low-level counties within county towns, county-level cities, suburbs and inner cities. (**b**): Proportions of high-, middle- and low-level counties within 13 prefecture-level cities in Beijing-Tianjin-Hebei urban agglomeration.

**Figure 9 ijerph-20-04429-f009:**
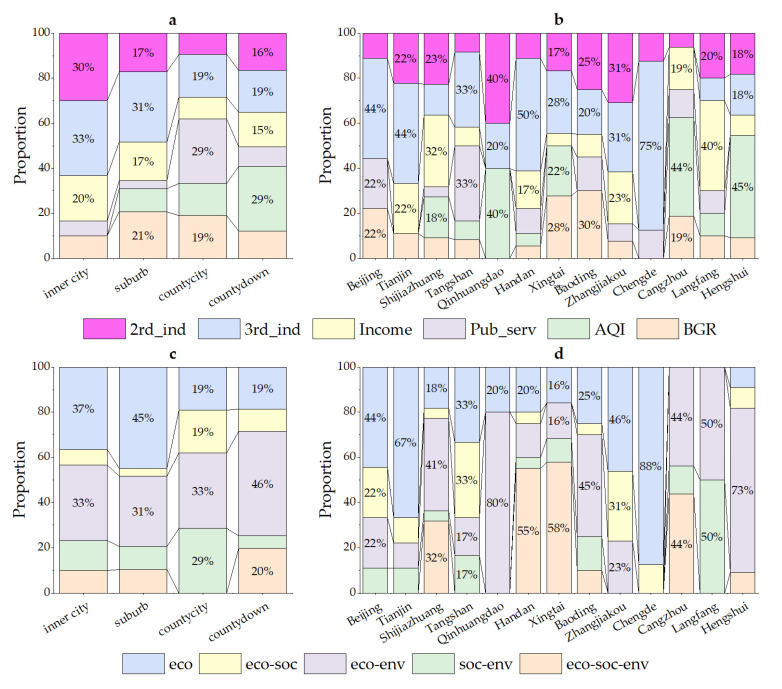
(**a**) Proportions of medium- and short-term improvement elements in four types of counties (inner cities, suburbs, county-level cities and county towns). (**b**) Proportions of medium- and short-term improvement elements within 13 prefecture-level cities. (**c**) Proportions of medium- and long-term improvement elements in four types of counties. (**d**) Proportions of medium- and long-term improvement elements within 13 prefecture-level cities.

**Figure 10 ijerph-20-04429-f010:**
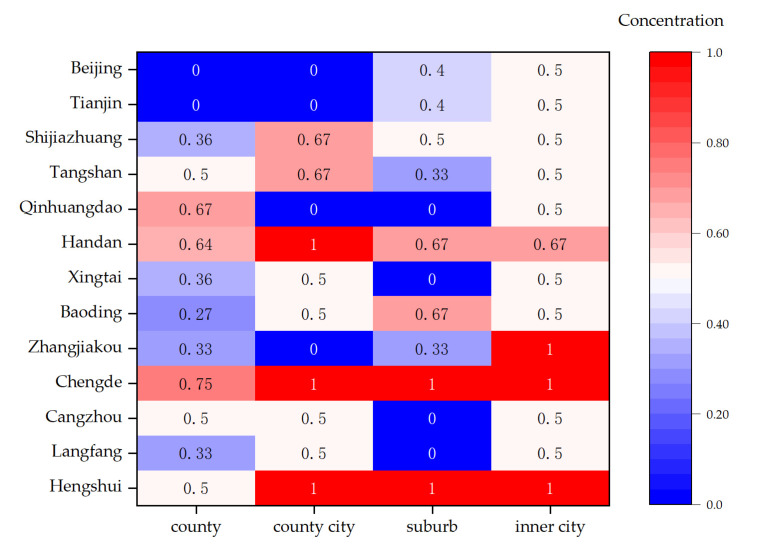
Concentration of short-term improvement elements in inefficient counties with different administrative types and located in various prefecture-level cities. Note: The concentration indicates the proportion of the dominant improvement elements in the unit, i.e., the ratio of the number of counties with that improvement element to the total number of counties in the unit. 2nd_ind—secondary industry; 3rd_ind—tertiary industry; Income—resident income; Pub_serv—public services; AQI—air quality; BGR—green development level.

**Table 1 ijerph-20-04429-t001:** Urban land use efficiency measurement indexes.

Types	Names	Details
Input index	Employees per land area	Number of employees in the secondary and tertiary industries/Construction land area
	Fixed asset investment per land area	Fixed asset investment of the whole society/Construction land area
Desirable output index	Added value of the secondary industry per land area	Added value of the secondary industry/Construction land area
	Added value of the tertiary industry per land area	Added value of the tertiary industry/Construction land area
	Per capita disposable income of urban residents	Per capita disposable income of urban residents
	Density of POI	Number of POI (medical care services, living facilities, science and education cultural services)/Construction land area
	Green coverage rate in built-up area	Greenland area/Construction land area
Undesirable output index	Concentration of PM2.5	Annual average concentration of PM2.5 concentration

**Table 2 ijerph-20-04429-t002:** Paired-samples *t*-test results based on efficiency scores of the SBM and MinDS models.

Samples	Mean	N	*t*-Value	Sig.	Correlation	Sig.
SBM	0.364	197	−38.009	0.001	0.822	0.001
MinDS	0.731					

Note: N—number of samples.

**Table 3 ijerph-20-04429-t003:** Homogeneity of variance and F-value test of global progress values of counties at different levels.

Test Variable	Classification	Levene Statistic	F-Value
Global progress value	2nd–7th level	1.709	7.524 ***

Note: *** show significance at the 1% level, respectively.

**Table 4 ijerph-20-04429-t004:** Homogeneity of variance and F-value test of local progress values of counties at different levels.

Test Variable	Classification	Mean	Levene Statistic	F Value
Local progress value (1st step)	2nd–7th level	1.436	1.178	0.856
Local progress value (2nd step)	3rd–7th level	1.367	1.359	1.554
Local progress value (3rd step)	5th–7th level	1.321	1.530	1.622

**Table 5 ijerph-20-04429-t005:** Paired-samples *t*-test results for progress values of the one-step and the step-by-step contexts.

Samples	Mean	N	*t*-Value	Sig.	Correlation	Sig.
step-by-step	1.327	184	−12.913	0.001	0.826	0.001
one step	1.504					

Note: N—number of samples.

## Data Availability

The data that support the findings of this study are available from the corresponding author upon reasonable request.

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
