# Peer review of "Improvement Pathways for Urban Land Use Efficiency in the Beijing-Tianjin-Hebei Urban Agglomeration at the County Level: A Context-Dependent DEA Based on the Closest Target"

_ijerph, 2023, doi:10.3390/ijerph20054429_

Round 1

Reviewer 1 Report

This paper seems to be a rather standard urban development/administration paper based on an elaboration of sophisticated economic models and is relevant for publication. However, the connection to this journal is tangential at best. The literature is largely related to land use and land use efficiency. More connection could presumably be made using the topic to the environment and public health, but I did not see it, other than generic and implicit connections that could be made to about anything. I would strongly suggest that this be submitted to a journal more directly related to the topic. It is clearly part of an informed academic conversation, just not one directly related to environment and public health. Perhaps something like Urban Affairs, Urban Affairs Review, Habitat International, or something along those lines.

If it is accepted here, it would require significant revision in order to establish much clearer implications for environmental quality and/or public health. I think the much easier path is to submit it to a more appropriate journal.

In addition, the paper employs large amounts of first person, making it sound casual, like for a seminar presentation. It would sound much more authoritative and less arrogant if it were written consistently in third person.

There are also occasional minor English mistakes in number and tense.

Reviewer 2 Report

The paper takes the land use efficiency of urban agglomeration area as the theme, and takes the Beijing-Tianjin-Hebei urban agglomeration as a case study. This study has practical significance for promoting the sustainable development of cities and promoting the transformation of urban land use mode from rough to intensive use. First of all, the paper uses DEA model to establish an indicator system to calculate the urban land use efficiency of each region. The research method is more scientific and the scheme is more mature. Secondly, the article focuses on empirical research, which transcends the theoretical orientation of traditional land use efficiency research, improves the practical value and application exploration of the research, especially the research focusing on the county level, and makes up for the lack of previous research focusing on the land use efficiency of large and medium-sized cities. Thirdly, the gradual improvement strategy of land use efficiency improvement based on the closest target of the county units with different land use efficiency proposed in the article has some new ideas.

The article has the following problems:

    First, in the evaluation index system of urban land use efficiency, the land use efficiency should be comprehensive efficiency (as described in the paper), including economic, social, ecological and other aspects, but as far as the current indicator system is concerned, the social and ecological aspects are not reflected enough (actually trade and environmental indicators). Secondly, the article puts forward improvement strategies for various units in terms of improving urban land use in the county, but the Beijing-Tianjin-Hebei region has a large difference, which is reflected in the economic scale, functional orientation, development direction, leading industries and other aspects. Can you propose targeted strategies for improving the efficiency of land use in the county units.

Reviewer 3 Report

The subject matter of the article is interesting. However, I would like to suggest some additions to improve it before publication.

1. In the introduction section, the research question is not clear enough and needs to be clearly stated.

2. in the literature review, the two categories of current ULUE issues are not well related to the core issues of this paper's research and the literature review needs to be more focused.

3. when explaining the county units, further explanation of the theoretical significance for ULUE is necessary.

4. in the research methodology, although it is stated in the abstract that "by utilizing methods such as the significant difference test and system clustering analysis, the shortest path and steps to achieve efficiency were identified for inefficient counties", the relationship between this method and e shortest path is not clearly described in the text.

5. In the discussion section, in addition to showing consistency with existing research, it is necessary to explain the innovative conclusions and methods drawn from this paper and their significance; the policy implications section needs to suggest specific strategies for different types of spatial units and indicate which areas these policies point to.

6. The conclusion section needs to be more explicit and informative in answering the research questions raised in the INTRODUCTION.

7. other minor issues: the font in the picture needs to be unified, and the full name of the DMU needs to be written when it first appears.

Sincerely

Author Response

请参阅附件

Round 2

Reviewer 1 Report

The changes outlined in the letter are sufficient so long as changes of concern to other reviewers are accomplished satisfactorily. 

Reviewer 3 Report

The article has been significantly improved, but there are still some important issues to be addressed.

1.The abstract and introduction are relatively long and need to be greatly simplified.In particular, one of the paragraphs in this revision is almost 1 page. This is very difficult for the reader to follow.Similarly, there is non-essential information in the summary that can be condensed.

2.The author(s) still need to ensure consistency between different figures, including font size, image size, etc.

sincerely
